# Pre-injury activity predicts outcomes following distal radius fractures in patients age 60 and older

Rachel C. Hooper[1], Nina Zhou[2], Lu Wang[3‡], Melissa J. Shauver[4], Kevin C. Chung[5‡]*, for the WRIST Group[¶]

1 Hand Surgery Fellow, Section of Plastic Surgery, Department of Surgery, University of Michigan Medical School, Ann Arbor, MI, United States of America, 2 PhD Candidate, Department of Biostatistics, University of Michigan, Ann Arbor, MI, United States of America, 3 Associate Professor, Department of Biostatistics, University of Michigan, Ann Arbor, MI, United States of America, 4 Clinical Research Coordinator, Section of Plastic Surgery, Department of Surgery, University of Michigan Medical School, Ann Arbor, MI, United States of America, 5 Charles B. G. de Nancrede Professor of Surgery, Section of Plastic Surgery, Department of Surgery, University of Michigan Medical School, Ann Arbor, MI, United States of America

☉ These authors contributed equally to this work.
‡ These authors also contributed equally to this work.
¶ Membership of the WRIST Group is listed in the Acknowledgments.
* kecchung@med.umich.edu

**Data Availability Statement:** S3 Data. Data file.

**Funding:** Research reported in this publication was supported by the National Institute of Arthritis and Musculoskeletal and Skin Diseases and the

## Abstract

### Introduction

One out of every 5 elderly patients will suffer a distal radius fracture and these injuries are often related to poor bone health. Several surgical subspecialties have demonstrated that pre-injury activity level can impact patient outcomes. To determine the importance of physical activity, we examined the relationship between pre-injury activity and patient-reported and functional outcomes among fracture patients.

### Methods

This is a retrospective analysis of prospectively collected data from participants enrolled in the Wrist and Radius Injury Surgical Trial (WRIST) from April 10, 2012 to December 31, 2016. This study included 304 adults, 60 years or older with isolated unstable distal radius fractures; 187 were randomized to one of three surgical treatments and 117 opted for casting. Participants opting for surgery were randomized to receive volar locking plate, percutaneous pinning, or external fixation. Participants who chose not to have surgery were treated with casting. All participants were stratified prior to analysis into highly and less-active groups based on pre-injury Rapid Assessment of Physical Activity Scores.

### Results

280 patients had 12-month assessments of outcomes. Highly active participants scored 8 and 5 points greater on the Michigan Hand Questionnaire at 6 weeks and 3 months respectively, p<0.05. Highly active participants demonstrated greater grip strength at the 3-month (p = 0.017) and 6-month (p = 0.007) time-points. Highly active participants treated with volar

National Institute on Aging of the National Institutes of Health under Award Number R01 AR062066 and by the National Institute of Arthritis and Musculoskeletal and Skin Diseases of the National Institutes of Health under Award Number 2 K24-AR053120-06.

**Competing interests:** No authors have competing interests.

locking plate scored 10+ points greater on the Michigan Hand Questionnaire compared to the less-active group at the 6-week (p = 0.032), 3-month (p = 0.009) and 12-month (0.004) time points, with an effect size larger than 0.50, suggesting pre-injury level of activity had a significant clinical impact.

## Conclusions

Higher levels of pre-injury activity are predictive of patient-reported and functional outcomes following distal radius fracture. Because of the greater PROs, the early mobility and lower risk of hardware infection reported in the literature, volar plating is preferable to other treatments for highly active patients who request and meet indications for surgery.

## Trial registration

clinicaltrials.gov identifier: NCT01589692.

## Introduction

Distal radius fractures (DRF) are among the most commonly encountered fractures and affect approximately 18% of older adults. [1–7] Elderly women are at greatest risk of fragility fractures with the decrease in estrogen levels and higher rate of osteoporosis. [7–9] Treatment after DRF includes consideration of a patient's lifestyle, co-morbidities, fracture stability, surgeon training and procedural expertise, and hospital setting and resources. [10] Much of the DRF literature focuses on fracture pattern, age and invasiveness of surgery to determine who should receive which treatment. Casting is reserved for low-demand, less-active patients or those patients who do not wish to undergo surgery. Surgical reduction and fixation, typically with a volar locking plate system (VLPS) is utilized in high-demand, active patients. [1,3–5]

Wrist fractures in older individuals with osteoporosis have multiple deleterious effects including increased mortality, reduced mobility, decline in physical function, and higher healthcare costs. [9–12] Aerobic and weight-bearing exercise are the most widely accepted, cost-effective means of improving bone health in older patients and are essential to fracture prevention and recovery. [8,13–14] Because sustained physical activity increases bone mineral density and attenuates bone loss, the National Osteoporosis Foundation recommends that women participate in weight-bearing exercise to prevent osteoporosis and fragility fractures. [8–11] Several studies have demonstrated a direct relationship between pre-injury level of physical activity and improved postoperative outcomes in cardiac, breast, hernia, gastrointestinal and orthopedic surgery. [15–22] Medicare recognizes the importance of exercise in preventing and treating several chronic conditions including diabetes and peripheral vascular disease. [23–24] Furthermore, Medicare Part C offers specific exercise benefits including "SilverSneakers" and "Silver and Fit" which partners with thousands of fitness centers across the country to offer supervised exercise, strength and training programs for older individuals. [25]

Related to advances in medicine and the success of programs like SilverSneakers, surgeons are encountering older patients who are increasingly active and independent. For indicated fracture types, this growing group of active patients may wish to undergo surgical treatment despite their age because this could minimize the period of immobilization and lead to earlier return to activity. This study examines patient-reported and functional outcomes in highly and less-active older participants following DRF. We assessed the relationship between pre-

injury level of physical activity and post-DRF treatment outcomes after casting, VLPS, closed reduction and percutaneous pinning (CRPP), and external fixator placement. We hypothesized that highly active participants will have better patient-reported and functional outcomes.

## Methods

### Study cohort

We performed a retrospective analysis of prospectively collected data from the Wrist and Radius Injury Surgical Trial (WRIST). Participants were DRF patients enrolled at 24 sites in the US, Canada, and Singapore. Inclusion criteria included age 60 years or older with an unstable fracture where surgery was the recommended treatment (dorsal angulation >10˚, radial inclination <15˚, or radial shortening >3mm). Surgical participants were randomized to receive internal fixation with VLPS, closed reduction and external fixation, or CRPP. Participants who did not want surgery were treated with casting and followed as an observation group. Exclusion criteria included nursing home residents or residents of other institutional settings, dementia, open or bilateral fractures, previous DRF to same wrist, and comorbid conditions prohibiting surgery.

Because previous studies in the literature have demonstrated similar functional outcomes following surgical and non-surgical treatment of DRF, we stratified participants based on the Rapid Assessment of Physical Activity (RAPA) at the time of enrollment. RAPA is a 9-item questionnaire developed for use among patients > 50 years old, based on recommendations from the Centers for Disease Control and Prevention regarding the appropriate amount of exercise necessary to decrease falls in this group. Responses are scored and patients are categorized as 1 = sedentary, 2 = underactive and 3 = active. Using this validated questionnaire, we derived two groups for our study: patients who scored 1 or 2 were categorized as "less-active" and those that scored 3 were categorized as "highly active." [13,26]

Participant assessments took place at 2 weeks, 6 weeks, 3, 6, and 12 months following final fracture reduction or surgery. Patient-reported outcomes included the Short-Form 36 (SF-36) and MHQ summary scores. SF-36 and MHQ were chosen as they are validated assessments of overall health and hand-specific disability respectively. [2,27,28] Functional outcomes included assessment of grip strength, wrist and forearm arc of motion at 6 weeks, 3, 6, and 12 months among the two groups. MHQ assessments were performed at the 6-week, 3-, 6- and 12-month time-points, whereas SF-36 was performed at the time of enrollment, 3-, 6-, and 12-months. The WRIST protocol was approved by institutional review boards at all sites. Written informed consent was obtained from all WRIST participants. A Data Safety and Monitoring Board appointed by the National Institute of Arthritis and Musculoskeletal and Skin Diseases oversaw the study.

### Statistical methods

The primary outcome was the MHQ summary score between highly- and less-active participants. Secondary outcomes included SF-36 score, grip strength, wrist and forearm arc of motion. To determine the appropriateness of comparing casted and surgical patients, we performed a statistical analysis of the demographic characteristics of the respective groups. Descriptive statistics were computed for the entire study cohort and for each activity level group separately, including mean and standard deviation for continuous variables, and frequency and percentage for categorical variables. Unadjusted between-group comparisons were conducted using two-sample t-tests or Fisher's exact tests to evaluate the group mean difference of each demographic variable. Mean outcome scores over time stratified by treatment

were plotted for highly and less-active participants to visually examine recovery trends in each treatment group and physical activity combination.

Two sample t-tests and multivariate linear models adjusting for demographic variables were performed to provide adjusted comparisons between activity groups on the outcomes of interest at each time-point. To confirm the difference at each time-point was clinically significant, we calculated an effect size that was derived using the estimated adjusted mean difference of each patient-reported or functional outcome measure between highly and less-active participants and standard deviation of the outcome measure among the entire group. [27] Effect sizes are classified as small (0.20), medium (0.50), large (0.80) and very large (1.20). [29–31] Because each participant had longitudinal data collected, we examined the association between activity level and MHQ summary score, grip strength, wrist and forearm arc of motion among participants who underwent casting, VLPS, CRPP, and external fixation treatments separately using *Generalized Estimating Equations* (GEE) with unstructured correlation structure to account for correlated repeated outcomes that are not normally distributed. GEE down-weighs redundant information among highly correlated outcomes from one individual so that they have a cluster effect toward the association between activity level and health outcomes. Time is coded as a categorical variable with reference cell coding (reference group: 6 weeks). The GEE model for each health outcome is specified as:

$$E(Outcome_{ij}) = \beta_0 + \beta_1 I(\text{patient i is highly active}) + \beta_2(\text{Ipatient i is highly active}) \times I(time_{ij} =$$
$$3 \text{ months}) + \beta_3 I(\text{patient i is highly active}) \times I(time_{ij} =$$
$$6 \text{ months}) + \beta_4 I(\text{patient is highly active}) \times I(time_{ij} = 12 \text{ months}) + demographics_i,$$

where i indicates the patient and j indicates time points including 6 weeks, 3 months, 6 months and 12 months. We calculated the adjusted mean differences in patient health care outcomes and corresponding effect sizes at different time points between the highly and less-active participants based on surgical treatment. The Wald test was utilized to derive the p-values for group differences at each time point.

## Results

A total of 280 participants had at least one observation for outcomes of interest during the 12-month study period. Two participants had missing RAPA scores and were excluded from the analysis. Comparison between the randomized surgical groups and non-randomized casting group revealed age (68 vs. 76 years, p<0.001) and race (p = 0.008) as the only significant demographic differences. (Table 1). There were no significant differences in sex, level of education, co-morbidities, smoking, employment status, or income (Table 1). Because there were minor demographic differences between the casted and non-casted group, we felt it was appropriate to compare all participants based on pre-injury activity level.

After stratification of patients based on RAPA score, 110 participants were classified as highly active and 170 were classified as less active (Table 2). A greater proportion of participants in the less-active group (42%) received casting compared with the highly active group (26%), p<0.05 (Table 2). On average, highly active participants were younger than less active participants, (68 vs. 73 years, p<0.001). 67% of highly active participants had some college or professional education (p = 0.03) and 40% of them earned $60,000 or more (p<0.01). Less active participants had a higher rate of hypertension (p = 0.03) and chronic obstructive pulmonary disease (p = 0.03). Both activity level groups were similar in terms of race, diabetes, smoking and employment status.

Highly active participants demonstrated (p<0.05) greater MHQ summary scores at all time-points. This trend remained significant at the 6-week and 3-month time points with

**Table 1. Comparison of demographic characteristics among the surgical group (VLPS, ExFix, Pinning) and the casting group.**

| | Surgical Group (n = 180) | Casting Group (n = 100) | P-value |
|---|---|---|---|
| **Average Age at Enrollment** (mean (SD) | 68.48 (7.29) | 75.68 (9.81) | <0.001 |
| **Sex** Count (%) | | | |
| Male (1) | 22 (12.2) | 14 (14.0) | |
| Female (2) | 158 (87.8) | 86 (86.0) | 0.811 |
| **Race** count (%) | | | 0.008 |
| American Indian/Alaskan Native | 1 (0.6) | 0 (0.0) | |
| Asian | 6 (3.3) | 15 (15.0) | |
| Pacific Islander/ Hawaii Native | 0 (0.0) | 1 (1.0) | |
| Black | 11 (6.1) | 5 (5.0) | |
| White | 159 (88.3) | 77 (77.0) | |
| 2+ or other | 1 (0.6) | 2 (2.0) | |
| Missing (NA) | 2 (1.1) | 0 (0.0) | |
| **Highest Level of Education** count (%) | | | 0.228 |
| <HS graduate | 19 (10.6) | 19 (19.0) | |
| HS diploma/GED | 39 (21.7) | 24 (24.0) | |
| Vocational/Technical School | 13 (7.2) | 2 (2.0) | |
| Some college/Associate | 44 (24.4) | 25 (25.0) | |
| College Graduate | 27 (15.0) | 13 (13.0) | |
| Professional | 33 (18.3) | 16 (16.0) | |
| Missing (NA) | 5 (2.8) | 1 (1.0) | |
| **Comorbidities** count (%) | | | |
| *Hypertension* | | | 0.171 |
| No | 89 (49.4) | 39 (39.0) | |
| Yes | 90 (50.0) | 61 (61.0) | |
| Missing (NA) | 1 (0.6) | 0 (0.0) | |
| *Diabetes* | | | 0.749 |
| No | 155 (86.1) | 86 (86.0) | |
| Yes | 24 (13.3) | 14 (14.0) | |
| Missing (NA) | 1 (0.6) | 0 (0.0) | |
| *COPD* | | | 0.647 |
| No | 163 (90.6) | 89 (89.0) | |
| Yes | 16 (8.9) | 11 (11.0) | |
| Missing (NA) | 1 (0.6) | 0 (0.0) | |
| **Smoking** count (%) | | | 0.546 |
| Never | 95 (52.8) | 52 (52.0) | |
| Current smoker | 18 (10.0) | 8 (8.0) | |
| Former smoker <10 years | 9 (5.0) | 2 (2.0) | |
| Former smoker >10 years | 57 (31.7) | 38 (38.0) | |
| Missing (NA) | 1 (0.6) | 0 (0.0) | |
| **Employment Status** count (%) | | | 0.270 |
| Full-time | 36 (20.0) | 9 (9.0) | |
| Part-time | 22 (12.2) | 12 (12.0) | |
| Retired | 109 (60.6) | 72 (72.0) | |
| Disability | 5 (2.8) | 2 (2.0) | |
| Full-time student | 0 (0.0) | 0 (0.0) | |
| Part-time student | 0 (0.0) | 0 (0.0) | |
| Unemployed | 6 (3.3) | 4 (4.0) | |

(*Continued*)

**Table 1.** (Continued)

| | Surgical Group (n = 180) | Casting Group (n = 100) | P-value |
|---|---|---|---|
| Missing (NA) | 2 (1.1) | 1 (1.0) | |
| **Income** count (%) | | | 0.114 |
| <$10K | 9 (5.0) | 9 (9.0) | |
| $10K-$59999 | 94 (52.2) | 61 (61.0) | |
| >$60K | 58 (32.2) | 20 (20.0) | |
| Missing (NA) | 19 (10.6) | 10 (10.0) | |

highly active patients scoring on average 8 (p<0.01) and 5 (p<0.05) points greater at the respective time points, after controlling for demographic variables including surgery treatment type, age at enrollment, sex, race, highest level of education, co-morbidities, smoking, employment status, and income (Table 3). The effect size of differences between the two groups was 0.40 and 0.30 at the 6-week and 3-month time-points suggesting these differences represents a small effect.

Highly active patients reported greater mean SF-36 physical scores at all examined time-points, p<0.01. After controlling for all other demographic variables, highly active participants on average scored 4, 6, 7, and 7 points greater on the SF-36 physical domain compared to the less active participants at enrollment and 3, 6, and 12 months respectively, p<0.001 (Table 3). A similar trend was noted in effect size; at baseline, 3-,6-, and 12-month time-points where the effect size was 0.30, 0.60, 0.50 and 0.50 at each time point respectively, suggesting these discernible differences have medium clinical significance and implications.

Highly active participants demonstrated significantly greater grip strength at the 3-month (p = 0.017) and 6-month (p = 0.007) time points when compared to less-active participants. The effect size of these differences was 0.30 and 0.40 at 3 months and 6 months respectively, demonstrating that the observed differences are of small clinical magnitude. Examination of wrist and forearm arc of motion revealed no significant functional outcome difference between the two activity groups.

The overall trend in recovery stratified by treatment group demonstrates that the rate of recovery is similar for both less and highly active groups; there were no significant differences in the recovery trend among VLPS highly and less active patients in wrist and forearm arc of rotation, or grip strength. (Fig 1). Whereas the absolute mean of the MHQ summary scores for highly active participants were greater at all time-points, the rate of MHQ score increase over time was similar among the highly and less active groups that underwent CRPP. Although the change is similar, greater patient-reported outcomes *earlier* in the recovery process among highly active patients over age 60, especially those who undergo VLPS is an important consideration during consultation with these patients to help them make a decision for or against surgery.

Table 4 compares patient-reported and functional outcomes between the highly and less-active participants in each treatment group over time. Highly active participants who underwent VLPS demonstrated a 10 to 14-point improvement on the MHQ questionnaire over less-active patients with medium effect size at 6 weeks (p = 0.032) and 3 (p = 0.009), and 12 months (p = 0.004) respectively. Highly active participants in the CRPP group scored 24, 13, 9, and 12 points greater on the MHQ assessment at 6 weeks and 3, 6, and 12 months respectively, p<0.001 (Table 4). With the exception of the 6-month time point, the effect size of these differences in MHQ scores between the groups were medium to large. Additionally, highly active participants who underwent CRPP demonstrated a 9 to 11 points greater SF-36 physical questionnaire score at all time points (p < 0.01) with large effect size (Table 4). Among participants

**Table 2. Demographic characteristics of highly and less-active patients.**

| | Overall (n = 280) | Less Active (n = 170, 61%) | Highly Active (n = 110, 39%) | P-value |
|---|---|---|---|---|
| **Treatment** count (% of all patients in group) | | | | 0.072 |
| VLPS | 63 (22.5) | 34 (20.0) | 29 (26.4) | 0.272 |
| Ex-Fix | 62 (22.1) | 35 (20.6) | 27 (24.5) | 0.528 |
| Pinning | 55 (19.6) | 30 (17.6) | 25 (22.7) | 0.373 |
| Casting | 100 (35.7) | 71 (41.8) | 29 (26.4) | 0.012 |
| **Average Age at Enrollment** mean (SD) | 71.05 (8.95) | 72.9 (9.08) | 68.2 (8.01) | <0.001 |
| **Sex** count (%) | | | | 0.220 |
| Male (1) | 36 (12.9) | 18 (10.6) | 18 (16.4) | |
| Female (2) | 244 (87.1) | 152 (89.4) | 92 (83.6) | |
| **Race** count (%) | | | | 0.138 |
| American Indian/Alaskan Native | 1 (0.4) | 0 (0.0) | 1 (0.9) | |
| Asian | 21 (7.5) | 17 (10.0) | 4 (3.6) | |
| Pacific Islander/ Hawaii Native | 1 (0.4) | 1 (0.6) | 0 (0.0) | |
| Black | 16 (5.7) | 9 (5.3) | 7 (6.4) | |
| White | 236 (84.3) | 138 (81.2) | 98 (89.1) | |
| 2+ or other | 3 (1.1) | 3 (1.8) | 0 (0.0) | |
| Missing (NA) | 2 (0.7) | 2 (1.2) | 0 (0.0) | |
| **Highest Level of Education** count (%) | | | | 0.027 |
| <HS graduate | 38 (13.6) | 30 (17.6) | 8 (7.3) | |
| HS diploma/GED | 63 (22.5) | 43 (25.3) | 20 (18.2) | |
| Vocational/Technical School | 15 (5.4) | 9 (5.3) | 6 (5.5) | |
| Some college/Associate | 69 (24.6) | 39 (22.9) | 30 (27.3) | |
| College Graduate | 40 (14.3) | 23 (13.5) | 17 (15.5) | |
| Professional | 49 (17.5) | 22 (12.9) | 27 (24.5) | |
| Missing (NA) | 6 (2.1) | 4 (2.4) | 2 (1.8) | |
| **Comorbidities** count (%) | | | | |
| *Hypertension* | | | | 0.026 |
| No | 128 (45.7) | 68 (40.0) | 60 (54.5) | |
| Yes | 151 (53.9) | 101 (59.4) | 50 (45.5) | |
| Missing (NA) | 1 (0.4) | 1 (0.6) | 0 (0.0) | |
| *Diabetes* | | | | 0.214 |
| No | 241 (86.1) | 142 (83.5) | 99 (90.0) | |
| Yes | 38 (13.6) | 27 (15.9) | 11 (10.0) | |
| Missing (NA) | 1 (0.4) | 1 (0.6) | 0 (0.0) | |
| *COPD* | | | | 0.033 |
| No | 252 (90.0) | 147 (86.5) | 105 (95.5) | |
| Yes | 27 (9.6) | 22 (12.9) | 5 (4.5) | |
| Missing (NA) | 1 (0.4) | 1 (0.6) | 0 (0.0) | |
| **Smoking** count (%) | | | | 0.701 |
| Never | 147 (52.5) | 85 (50.0) | 62 (56.4) | |
| Current smoker | 26 (9.3) | 18 (10.6) | 8 (7.3) | |
| Former smoker <10 years | 11 (3.9) | 7 (4.1) | 4 (3.6) | |
| Former smoker >10 years | 95 (33.9) | 59 (34.7) | 36 (32.7) | |
| Missing (NA) | 1 (0.4) | 1 (0.6) | 0 (0.0) | |
| **Employment Status** count (%) | | | | 0.094 |
| Full-time | 45 (16.1) | 20 (11.8) | 25 (22.7) | |
| Part-time | 34 (12.1) | 20 (11.8) | 14 (12.7) | |

(*Continued*)

**Table 2.** (Continued)

| | Overall (n = 280) | Less Active (n = 170, 61%) | Highly Active (n = 110, 39%) | P-value |
|---|---|---|---|---|
| Retired | 181 (64.6) | 120 (70.6) | 61 (55.5) | |
| Disability | 7 (2.5) | 4 (2.4) | 3 (2.7) | |
| Full-time student | 0 (0.0) | 0 (0.0) | 0 (0.0) | |
| Part-time student | 0 (0.0) | 0 (0.0) | 0 (0.0) | |
| Unemployed | 10 (3.6) | 5 (2.9) | 5 (4.5) | |
| Missing (NA) | 3 (1.1) | 1 (0.6) | 2 (1.8) | |
| **Income** | | | | 0.001 |
| <$10K | 18 (6.4) | 14 (8.2) | 4 (3.6) | |
| $10K-$59999 | 155 (55.4) | 103 (60.6) | 52 (47.3) | |
| >$60K | 78 (27.9) | 34 (20.0) | 44 (40.0) | |
| Missing (NA) | 29 (10.4) | 19 (11.2) | 10 (9.1) | |

treated with casting there were no significant differences in MHQ scores between the groups; however, highly active participants demonstrated a greater grip strength at all time points with small to medium effect sizes (Table 4).

## Discussion

By 2030, the US Census projects that persons over 65 years of age will outnumber children for the first time in US history. [32] DRFs are the second most common fracture in the elderly and an estimated 18% of the growing older population stand to suffer this fracture. [9,12] Exercise increases bone mineral density and functional adaptation in response to loading. [8] Wrist fractures are often the gateway to subsequent fragility fractures including hip and vertebral fractures and much attention has been given to the prevention and treatment of these fractures. [12] Most of the previous DRF literature examines the impact of age, treatment, and therapy on outcomes. [1–7, 27] Ikpeze et al, reported that women who suffered a DRF experienced a 50% functional decline compared to uninjured women. [9] Hakestad et al, compared postmenopausal women with low bone mineral density who suffered DRFs to uninjured healthy age-matched controls and found DRF patients with low bone mineral density had poor quality of life, decreased dynamic balance and physical capacity compared to controls. [12] Because there is an increasing number of older patients at risk of suffering a DRF, it is prudent that we devote attention to prevention and treatment strategies to facilitate quick and safe return to baseline function and high quality of life.

"Prehabilitation" or "training for surgery" has been widely adopted in other surgical subspecialties. [15–20] Within the DRF literature, little is known about how level of pre-injury activity influences functional outcomes. [1,4–5,9,20,33,27] In addition to age, co-morbidities, occupation, and fracture pattern/geometry, activity level is an important consideration in the DRF treatment algorithm. The current study demonstrates that increased pre-injury activity level has a positive impact on patient-reported and functional outcomes. Highly active participants had greater grip strength at all times points with a medium effect size. Additionally, highly active patients in the VLPS treatment group had greater MHQ summary scores at the 6-week time-point; among CRPP participants, greater MHQ scores were demonstrated at all time-points with a corresponding high effect size. Among VLPS patients, there were no significant functional outcomes between highly and less active patients. Active CRPP participants had significantly greater MHQ summary scores at the 6-week time point. Although highly active participants treated with CRPP had greater patient-reported and functional outcomes,

**Table 3. Patient-reported and functional outcome comparisons between highly and less-active participants.**

| Health Outcomes | Highly Active (n = 112,38.1%) Mean (SD) | Less Active (n = 182,61.9%) Mean (SD) | Adjusted Difference (P-value) | Adjusted Difference Effect Size | Adjusted Change from 6 Week (P-value) |
|---|---|---|---|---|---|
| **MHQ summary score** | | | | | |
| 6 weeks | 51 (20.68) | 42 (18.30) | 7.80 (0.003) | 0.40 | |
| 3 months | 71 (19.29) | 65 (19.66) | 5.20 (0.051) | 0.26 | -2.19 (0.379) |
| 6 months | 79 (19.58) | 75 (18.68) | 1.19 (0.650) | 0.06 | -7.94 (0.005) |
| 12 months | 86 (16.40) | 80 (18.13) | 2.75 (0.324) | 0.16 | -5.81 (0.082) |
| **SF-36 Physical** | | | | | |
| Baseline | 37 (9.92) | 33 (10.13) | 2.93 (0.019) | 0.28 | |
| 3 months | 48 (8.36) | 42 (9.69) | 5.50 (<0.001) | 0.57 | 2.86 (0.047) |
| 6 months | 50 (8.81) | 43 (11.07) | 5.68 (<0.001) | 0.52 | 2.79 (0.079) |
| 12 months | 50 (9.57) | 43 (11.38) | 5.31 (<0.001) | 0.47 | 3.48 (0.032) |
| **SF-36 Mental** | | | | | |
| Baseline | 51 (13.15) | 49 (14.09) | 1.05 (0.561) | 0.08 | |
| 3 months | 55 (9.25) | 54 (11.01) | 0.05 (0.975) | 0.01 | -1.16 (0.524) |
| 6 months | 54 (9.21) | 55 (9.45) | -1.37 (0.328) | -0.15 | -3.29 (0.100) |
| 12 months | 55 (8.40) | 54 (10.43) | 0.52 (0.733) | 0.05 | -0.50 (0.814) |
| **Wrist Arc of Motion (degrees)** | | | | | |
| 6 weeks | 56 (27.89) | 60 (29.36) | -5.26 (0.241) | -0.18 | |
| 3 months | 90 (24.31) | 88 (27.11) | 0.20 (0.959) | 0.01 | 2.68 (0.594) |
| 6 months | 108 (22.82) | 102 (25.47) | 4.23 (0.284) | 0.17 | 2.51 (0.674) |
| 12 months | 115 (20.41) | 106 (26.53) | 6.35 (0.111) | 0.26 | 9.74 (0.148) |
| **Forearm Arc of Motion** | | | | | |
| 6 weeks | 129 (36.88) | 125 (44.24) | 2.73 (0.656) | 0.07 | |
| 3 months | 152 (23.25) | 152 (27.11) | 0.65 (0.861) | 0.03 | -3.85 (0.501) |
| 6 months | 161 (18.23) | 160 (25.40) | 0.80 (0.830) | 0.04 | -3.82 (0.622) |
| 12 months | 166 (20.41) | 167 (18.28) | -1.81 (0.544) | -0.10 | 10.95 (0.148) |
| **Grip strength** | | | | | |
| 6 weeks | 6 (5.12) | 4 (4.60) | 0.79 (0.289) | 0.16 | |
| 3 months | 11 (6.93) | 8 (5.31) | 2.00 (0.017) | 0.32 | 0.74 (0.302) |
| 6 months | 16 (7.40) | 11 (6.22) | 2.67 (0.007) | 0.38 | 0.54 (0.602) |
| 12 months | 18 (7.47) | 15 (7.02) | 1.46 (0.163) | 0.20 | 0.08 (0.949) |

Controlled demographics: treatment, age at enrollment, sex, race, highest level of education, co-morbidities (hypertension, diabetes, chronic obstructive pulmonary disease), smoking, employment status, and income. Effect sizes: small (0.20–0.50), medium (0.51–0.80), large (> 0.80).

the pin care requirements, infection, and pin migration in the literature *may* outweigh these benefits. [27] Because of the aforementioned factors, the majority of operative distal radius fractures are treated with volar plating. However, at their last meeting, the American Academy of Orthopedic Surgery was unable to recommend for or against one method of fixation above another. [34] Thus, surgeons and patients must weigh the importance of early active range of motion, need for pin care, risk of hardware infection, as well as patient-reported outcomes in their ultimate decision for fixation of distal radius fractures.

There are some limitations to the current study. RAPA was determined by a self-reported questionnaire and may suffer from patient recall bias; however, the validity of patient-reported physical activity has been well-established and used routinely in research. [13–26,27] Bone

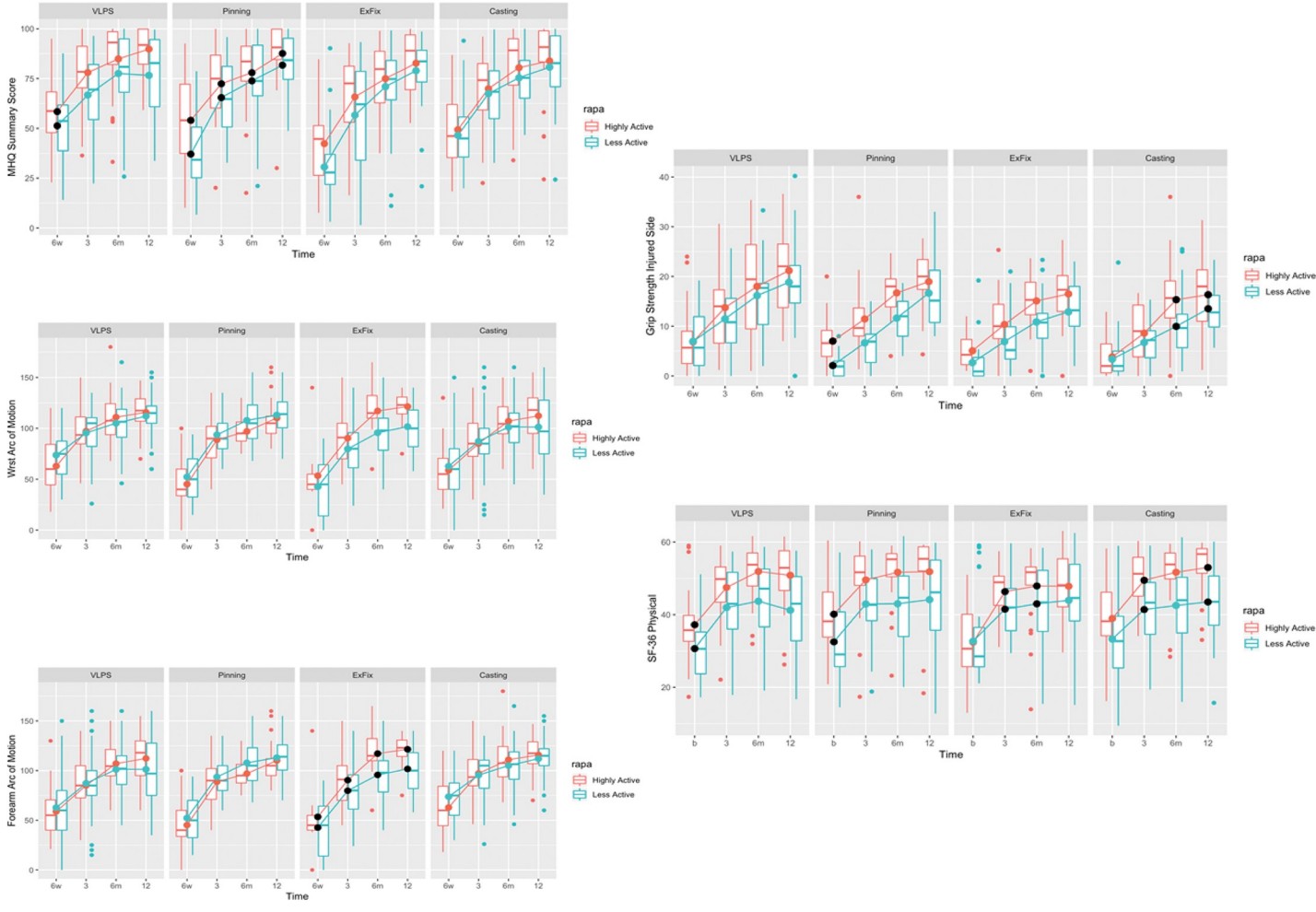

**Fig 1.**

mineral density and radiographic appearance of fracture sites were not specifically measured and would be of benefit to substantiate the mentioned benefits of weight bearing, high intensity exercises on healing and bone health. A greater proportion of less-active participants opted for casting and this is a potential confounder of the patient-reported and functional outcome differences; however, comparison of the casted and non-casted groups revealed no significant differences in medical co-morbidities.

## Conclusions

With later retirement, increased need for independence and demands for high quality treatment, surgeons must carefully determine which intervention an older DRF patient may warrant. Based on this study, higher levels of pre-injury activity are predictive of better patient-reported and functional outcomes. Because there is an increased risk of falls and fractures with more activity, supervised physical activity among the elderly is recommended. As surgeon proficiency grows with VLPS, we believe this method of fixation should be considered for DRF in highly active patients regardless of age given the improved patient-reported outcomes. Because casting produced comparable patient-reported outcomes among highly and less-active participants, we believe this is a suitable treatment for less active patients. We suggest surgeons

**Table 4. Generalized estimating equation model results comparing highly and less-active participants based on treatment.** The estimated mean differences and p-values are for $\beta_1$ (6 weeks), $\beta_1+\beta_2$ (3 months), $\beta_1+\beta_3$ (6 months) and $\beta_1+\beta_4$ (12 months), respectively.

| | Treatment Groups | | | | | | | |
|---|---|---|---|---|---|---|---|---|
| | VLPS | | Pinning | | ExFix | | Casting | |
| Health Outcomes | Estimated Mean Difference (P-value) | Effect Size | Estimated Mean Difference (P-value) | Effect Size | Estimated Mean Difference (P-value) | Effect Size | Estimated Mean Difference (P-value) | EffectSize |
| MHQ Summary Score | | | | | | | | |
| 6 weeks | 9.70 (0.032) | 0.52 | 24.27 (<0.001) | 1.13 | 10.14 (0.033) | 0.49 | -1.22 (0.770) | -0.07 |
| 3 months | 13.40 (0.009) | 0.67 | 12.95 (<0.001) | 0.66 | 7.52 (0.194) | 0.57 | -1.24 (0.117) | -0.08 |
| 6 months | 9.83 (0.079) | 0.49 | 9.27 (<0.001) | 0.46 | 2.77 (0.132) | 0.48 | -3.27 (0.541) | -0.20 |
| 12 months | 13.73 (0.004) | 0.75 | 12.19 (<0.001) | 0.78 | 4.88 (0.199) | 0.79 | -5.29 (0.992) | -0.28 |
| Wrist Arc of Motion | | | | | | | | |
| 6 weeks | -6.11 (0.449) | -0.24 | -1.58 (0.856) | -0.06 | 12.68 (0.250) | 0.40 | -7.39 (0.269) | -0.26 |
| 3 months | 5.56 (0.565) | 0.23 | -0.33 (0.033) | -0.02 | 6.90 (0.805) | 0.26 | -3.19 (0.916) | -0.12 |
| 6 months | 7.20 (0.613) | 0.30 | -5.22 (0.566) | -0.25 | 18.27 (0.118) | 0.65 | -1.13 (0.915) | -0.05 |
| 12 months | 5.65 (0.802) | 0.26 | 2.20 (0.132) | 0.10 | 22.03 (0.009) | 0.97 | 3.97 (0.752) | 0.13 |
| Forearm Arc of Motion | | | | | | | | |
| 6 weeks | -3.07 (0.747) | -0.08 | 12.80 (0.344) | 0.29 | 35.11 (0.009) | 0.66 | -11.02 (0.151) | -0.31 |
| 3 months | 0.95 (0.167) | 0.05 | -1.75 (0.895) | -0.08 | 0.16 (0.049) | 0.01 | 0.58 (0.344) | 0.02 |
| 6 months | 8.15 (0.752) | 0.29 | 4.02 (0.845) | 0.20 | -3.46 (0.039) | -0.23 | 2.01 (0.456) | 0.09 |
| 12 months | 1.40 (0.185) | 0.09 | 10.38 (0.182) | 0.74 | -5.58 (0.029) | -0.34 | -8.08 (0.535) | -0.35 |
| **SF-36 Physical** | | | | | | | | |
| 6 weeks | 5.86 (0.011) | 0.62 | 9.54 (0.003) | 0.88 | -2.85 (0.268) | -0.28 | 3.41 (0.045) | 0.33 |
| 3 months | 5.48 (0.056) | 0.54 | 9.11 (<0.001) | 0.89 | 3.12 (0.314) | 0.38 | 7.32 (0.001) | 0.73 |
| 6 months | 7.98 (0.006) | 0.75 | 11.43 (<0.001) | 1.02 | 3.50 (0.273) | 0.31 | 6.71 (0.002) | 0.62 |
| 12 months | 9.35 (<0.001) | 0.82 | 10.66 (<0.001) | 0.85 | 2.32 (0.532) | 0.21 | 7.41 (0.001) | 0.71 |
| Grip Strength | | | | | | | | |
| 6 weeks | -2.42 (0.154) | -0.41 | 5.25 (<0.001) | 1.14 | 1.47 (0.367) | 0.35 | -1.19 (0.277) | -0.30 |
| 3 month | -0.20 (0.383) | -0.03 | 4.60 (0.860) | 0.73 | 2.64 (0.854) | 0.44 | 1.08 (0.134) | 0.24 |
| 6 month | 0.30 (0.340) | 0.04 | 4.78 (0.094) | 0.91 | 3.29 (0.374) | 0.55 | 3.93 (0.750) | 0.57 |
| 12 month | 0.01 (0.875) | 0.00 | 2.84 (0.939) | 0.42 | 1.74 (0.407) | 0.28 | 4.87 (0.214) | 0.81 |

Controlled demographics: age at enrollment, sex, race, highest level of education, co-morbidities (hypertension, diabetes, chronic obstructive pulmonary disease), smoking, employment status, and income.

Effect sizes: small (0.20–0.50), medium (0.51–0.80), large (> 0.80).

continue to specifically incorporate activity level during pre-surgical evaluation and use activity level as a tool to guide patient treatment and predict outcomes.

## Supporting information

**S1 File. List of intuitional review boards and ethics committees involved in WRIST.**
(DOCX)

**S1 Data. Data file.**
(CSV)

## Acknowledgments

The WRIST Group—Michigan Medicine (Coordinating Center): Kevin C. Chung, MD, MS (Principal Investigator & Lead Author); H. Myra Kim, ScD (Study Biostatistician); Steven C.

Haase, MD; Jeffrey N. Lawton, MD; John R. Lien, MD; Adeyiza O. Momoh, MD; Kagan Ozer, MD; Erika D. Sears, MD, MS; Jennifer F. Waljee, MD, MPH; Matthew S. Brown, MD; Hoyune E. Cho, MD; Brett F. Michelotti, MD; Sunitha Malay, MPH (Study Coordinator); Melissa J. Shauver, MPH (Study Coordinator). Beth Israel Deaconess Medical Center: Tamara D. Rozental, MD (Co-Investigator); Paul T. Appleton, MD; Edward K. Rodriguez, MD, PhD; Laura N. Deschamps, DO; Lindsay Mattfolk, BA; Katiri Wagner. Brigham and Women's Hospital: Philip Blazar, MD (Co-Investigator); Brandon E. Earp, MD; W. Emerson Floyd; Dexter L. Louie, BS. Duke Health: Fraser J. Leversedge, MD (Co-Investigator); Marc J. Richard, MD; David, S. Ruch, MD; Suzanne Finley, CRC; Cameron Howe, CRC; Maria Manson; Janna Whitfield, BS. Fraser Health Authority: Bertrand H. Perey, MD (Co-Investigator); Kelly Apostle, MD, FRCSC; Dory Boyer, MD, FRCSC; Farhad Moola, MD, FRCSC; Trevor Stone, MD, FRCSC; Darius Viskontas, MD, FRCSC; Mauri Zomar, CCRP; Karyn Moon; Raely Moon. HealthPartners Institute for Education and Research: Loree K. Kalliainen, MD, MA (Co-Investigator, now at University of North Carolina Health Care); Christina M. Ward, MD (Co-Investigator); James W. Fletcher, MD; Cherrie A. Heinrich, MD; Katharine S. Pico, MD; Ashish Y. Mahajan, MD; Brian W. Hill, MD; Sandy Vang, BA. Johns Hopkins Medicine: Dawn M. Laporte, MD (Co-Investigator); Erik A. Hasenboehler, MD; Scott D. Lifchez, MD; Greg M. Osgood, MD; Babar Shafiq, MD, MS; Jaimie T. Shores, MD; Vaishali Laljani. Kettering Health Network: H. Brent Bamberger, DO (Co-Investigator); Timothy W. Harman, DO; David W. Martineau, MD; Carla Robinson, PA-C, MPAS; Brandi Palmer, MS, PC, CCRP. London Health Sciences Centre: Ruby Grewal, MD, MS (Co-Investigator); Ken A. Faber, MD; Joy C. MacDermid, PhD (Study Epidemiologist); Kate Kelly, MSc, MPH; Katrina Munro; Joshua I. Vincent, PT, PhD. Massachusetts General Hospital: David Ring, MD, PhD (Co-Investigator, now at University of Texas Health Austin); Jesse B. Jupiter, MD, MA; Abigail Finger, BA; Jillian S. Gruber, MD; Rajesh K. Reddy; Taylor M. Pong; Emily R. Thornton, BSc. Mayo Clinic: David G. Dennison, MD (Co-Investigator); Sanjeev Kakar, MD; Marco Rizzo, MD; Alexander Y. Shin, MD; Tyson L. Scrabeck, CCRP. The MetroHealth System: Kyle Chepla, MD (Co-Investigator); Kevin Malone, MD; Harry A. Hoyen, MD; Blaine Todd Bafus, MD; Roderick B. Jordan, MD; Bram Kaufman, MD; Ali Totonchil, MD; Dana R. Hromyak, BS, RRT; Lisa Humbert, RN. National University of Singapore: Sandeep Sebastin, MCh (Co-Investigator), Sally Tay. Northwell Health: Kate W. Nellans, MD, MPH (Co-Investigator); Sara L. Merwin, MPH. Norton Healthcare: Ethan W. Blackburn, MD (Co-Investigator); Sandra J. Hanlin, APRN, NP-C; Barbara Patterson, BSN, CCRC. OrthoCarolina Research Institute: R. Glenn Gaston, MD (Co-Investigator); R. Christopher Cadderdon, MD; Erika Gordon Gantt, MD; John S. Gaul, MD; Daniel R. Lewis, MD; Bryan J. Loeffler, MD; Lois K. Osier, MD; Paul C. Perlik, MD; W. Alan Ward, MD; Benjamin Connell, BA, CCRC; Pricilla Haug, BA, CCRC; Caleb Michalek, BS, CCRC. Pan Am Clinic/University of Manitoba: Tod A. Clark, MD, MSc, FRCSC(Co-Investigator); Sheila McRae, MSc, PhD. University of Connecticut Health: Jennifer Moriatis Wolf, MD (Co-Investigator, now at University of Chicago Medicine); Craig M. Rodner, MD; Katy Coyle, RN. University of Oklahoma Medicine: Thomas P. Lehman, MD, PT (Co-Investigator); Yuri C. Lansinger, MD; Gavin D. O'Mahony, MD; Kathy Carl, BA, CCRP; Janet Wells. University of Pennsylvania Health System: David J. Bozentka, MD (Co-Investigator); L. Scott Levin, MD; David P. Steinberg, MD; Annamarie D. Horan, PhD; Denise Knox, BS; Kara Napolitano, BS. University of Pittsburgh Medical Center: John Fowler, MD (Co-Investigator); Robert Goitz, MD; Cathy A. Naccarelli; Joelle Tighe. University of Rochester: Warren C. Hammert, MD, DDS (Co-Investigator); Allison W. McIntyre, MPH; Krista L. Noble; Kaili Waldrick. University of Washington Medicine: Jeffery B. Friedrich, MD (Co-Investigator); David Bowman; Angela Wilson. Wake Forest Baptist Health: Zhongyu Li, MD, PhD (Co-Investigator); L. Andrew Koman, MD; Benjamin R. Graves, MD; Beth P. Smith, PhD; Debra Bullard.

## Author Contributions

**Conceptualization:** Rachel C. Hooper, Kevin C. Chung.

**Data curation:** Lu Wang.

**Formal analysis:** Rachel C. Hooper, Nina Zhou, Lu Wang, Melissa J. Shauver, Kevin C. Chung.

**Funding acquisition:** Kevin C. Chung.

**Investigation:** Rachel C. Hooper, Melissa J. Shauver, Kevin C. Chung.

**Methodology:** Rachel C. Hooper, Nina Zhou, Lu Wang, Melissa J. Shauver, Kevin C. Chung.

**Project administration:** Rachel C. Hooper, Melissa J. Shauver.

**Supervision:** Lu Wang.

**Validation:** Lu Wang.

**Writing – original draft:** Rachel C. Hooper, Nina Zhou, Lu Wang, Melissa J. Shauver, Kevin C. Chung.

**Writing – review & editing:** Rachel C. Hooper, Nina Zhou, Lu Wang, Melissa J. Shauver, Kevin C. Chung.

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
