## [Decision Letter · Decision Letter 0]

24 Feb 2020

PONE-D-20-03237

Pre-Injury Activity Predicts Outcomes Following Distal Radius Fractures in Patients Age 60 and Older

PLOS ONE

Dear Dr. Hooper,

Thank you for submitting your manuscript to PLOS ONE. After careful consideration, we feel that it has merit but does not fully meet PLOS ONE’s publication criteria as it currently stands. Therefore, we invite you to submit a revised version of the manuscript that addresses the points raised during the review process.

The reviewers made an enormous effort to suggest improvements. Please give an answer to each note. A

We would appreciate receiving your revised manuscript by Apr 09 2020 11:59PM. To enhance the reproducibility of your results, we recommend that if applicable you deposit your laboratory protocols in protocols.io, where a protocol can be assigned its own identifier (DOI) such that it can be cited independently in the future. For instructions see: http://journals.plos.org/plosone/s/submission-guidelines#loc-laboratory-protocols

We look forward to receiving your revised manuscript.

Kind regards,

Hans-Peter Simmen, M.D., Professor of Surgery

Academic Editor

PLOS ONE

Journal Requirements:

2. Please provide additional details regarding participant consent. In the ethics statement in the Methods and online submission information, please ensure that you have specified (1) whether consent was suitably informed and (2) what type you obtained (for instance, written or verbal). If the need for consent was waived by the ethics committee, please include this information.

3. Thank you for your ethics statement : "The WRIST protocol was approved by institutional review boards at all sites."

5. One of the noted authors is a group or consortium [WRIST Group]. In addition to naming the author group, please list the individual authors and affiliations within this group in the acknowledgments section of your manuscript. Please also indicate clearly a lead author for this group along with a contact email address.

Reviewers' comments:

Reviewer's Responses to Questions

**Comments to the Author**

1. Is the manuscript technically sound, and do the data support the conclusions?

Reviewer #1: Yes

Reviewer #2: Yes

2. Has the statistical analysis been performed appropriately and rigorously? 

Reviewer #1: Yes

Reviewer #2: Yes

3. Have the authors made all data underlying the findings in their manuscript fully available?

Reviewer #1: Yes

Reviewer #2: Yes

4. Is the manuscript presented in an intelligible fashion and written in standard English?

Reviewer #1: Yes

Reviewer #2: Yes

5. Review Comments to the Author

Reviewer #1: The authors (including the WRIST group) presented their prospectively collected data about elderly patients with a distal radius fracture. Enrolled patients were stratified into highly and less-active groups based on an activity score. Next, patients were either randomized to one of three surgical treatment modalities or opted for casting. Highly active patients showed better patient-reported and functional outcomes than less-active patients during a 12-months-study period.

While the study and the manuscript are nicely written, I have some concerns:

Abstract:

Please make clear if the stratification into highly and less-active groups took place before randomization or before analysis.

I disagree with some conclusions and I would focus on your findings.

Methods:

Any other instability criteria?

Results:

I suggest to start this part with a positive sentence.

From the data (especially the figures), I get the impression that highly active patients start high in the scores and end up high. However, the difference (delta) between 6 weeks and 12 months seem independent of the activity level and the treatment modality. All improved about the same %. Please comment on this topic.

Patients after an operation usually get hand herapy. What kind of hand therapy did the enrolled patients receive?

Discussion:

I expect some limitations due to multicentricity and enrollment of American, Canadian, and Singapore patients.

Reviewer #2: The claim of this paper is to examine patient-reported and functional outcomes in highly and less-active older participants following distal radius fracture. Additionally the authors assessed the relationship between pre-injury level of physical activity and postoperative outcomes after casting, volar plating system, percutaneous pinning and external fixator placement. How significant is for the traumatology to know the pre-injury level of physical acitivity if the surgeon choose the percutaneous pinning or the external fixator methods?

In the current literature there are main reasons in operating elder patient with isolated radius fractures. On the one hand they have bad soft tissue conditions on the other hand you have to avoid a long anesthesia periode. The claims of the study are not properly placed in this context.

The date and analyses support the claims but in their conclusion the authors are saying casting is a suitable treatment for less active patients based on comparable patient-reported outcomes. But this decision making has to set in a context related to soft tissue damage or other contraindication for an operative treatment of distal radius fractures.

Abstract

Line 58: Volare plates cannot be used in all elderly patients. The above conclusion applies only to a selective patient population

Introduction

Line 70: Casting is reserved for low-demand, less-acitve patients…. There are also active patients who simply do not want surgery and therefore also qualify for casting treatments.

Methods

Line 99: The instability criteria listed are not complete. If this is wanted or there were no unstable radius fractures with termination of the palmar / dorsal joint lip, termination of the styloid ulnar process near the base, radioulnar dissociation, dorsal tilt of the peripheral fragment > 20 °, palmar tilt of the peripheral fragment > 20 °.

Results

Unfortunately, no tables were included in the manuscript for the reviewer. The table titles could only be read under numbers 358-363.

Discussion

Line 228/229: I think in the discussion a gradation of the criteria regarding the therapy algorithm should be worked out, whereby the preoperative activity level is weighted less than the fracture pattern

Line 242/243: If you are already performing an operative intervention, you should aim at least for an exercise-stable osteosynthesis in order to be able to carry out a functional after-treatment. Otherwise the operational risk of complications is too great.

Line 256/257: Even if the risk of falls in active patients surely increases, supervision cannot be required as a prerequisite for their activities. That would cause costs to rise massively

6. PLOS authors have the option to publish the peer review history of their article (what does this mean?). If published, this will include your full peer review and any attached files.

Reviewer #1: Yes: Valentin Neuhaus, MD

Reviewer #2: No

---

## [Author Response · Author response to Decision Letter 0]

5 Mar 2020

PONE-D-20-03237: Pre-Injury Activity Predicts Outcomes Following Distal Radius Fractures in Patients Age 60 and Older

Journal Requirements:

Response: We have revised the style to meet PLOS ONE requirements.

2. Please provide additional details regarding participant consent. In the ethics statement in the Methods and online submission information, please ensure that you have specified (1) whether consent was suitably informed and (2) what type you obtained (for instance, written or verbal). If the need for consent was waived by the ethics committee, please include this information.

Response: We have made this addition. 

Lines 195-196: “Written informed consent was obtained from all WRIST participants.”

3. Thank you for your ethics statement: "The WRIST protocol was approved by institutional review boards at all sites." Please amend your current ethics statement to include the full name of the ethics committee/institutional review board(s) that approved your specific study.

Response: Because there are 24 sites in WRIST we have made the list of ethics committees and institutional review board available in a supplemental file. 

S2 File. List of Intuitional Review Boards and Ethics Committees involved in WRIST.

Response: this has been done

5. In your Data Availability statement, you have not specified where the minimal data set underlying the results described in your manuscript can be found. PLOS defines a study's minimal data set as the underlying data used to reach the conclusions drawn in the manuscript and any additional data required to replicate the reported study findings in their entirety. Upon re-submitting your revised manuscript, please upload your study’s minimal underlying data set as either Supporting Information files or to a stable, public repository and include the relevant URLs, DOIs, or accession numbers within your revised cover letter.

Response: this has been done

6. One of the noted authors is a group or consortium [WRIST Group]. In addition to naming the author group, please list the individual authors and affiliations within this group in the acknowledgments section of your manuscript. Please also indicate clearly a lead author for this group along with a contact email address.

Response: We have moved the list of WRIST contributors to the acknowledgment section. 

Reviewer #1:

1. Abstract: Please make clear if the stratification into highly and less-active groups took place before randomization or before analysis. 

Response: The stratification took place before the analysis. This has been added, line 97. 

2. Methods: Any other instability criteria?

Response: These are the recommendations from the American Academy of Orthopedic Surgeons on the management of distal radius fractures and thus were used for the study. 

3. Results: I suggest to start this part with a positive sentence.

Response: This has been changed, lines 236-237. 

4. From the data (especially the figures), I get the impression that highly active patients start high in the scores and end up high. However, the difference (delta) between 6 weeks and 12 months seem independent of the activity level and the treatment modality. All improved about the same %. Please comment on this topic.

Response: 

We have clarified this point. 

Lines 281-285: “Although the change is similar, greater patient-reported outcomes earlier in the recovery process among highly-active patients over age 60, especially those who undergo VLPS (the most commonly used surgical treatment) is an important consideration during consultation with these patients to help them make a decision for or against surgery.” 

5. Patients after an operation usually get hand therapy. What kind of hand therapy did the enrolled patients receive?

Response: We did not standardize hand therapy. Participants received the standard of care therapy regimen at their treating institution. 

6. Discussion: I expect some limitations due to multicentricity and enrollment of American, Canadian, and Singapore patients.

Response: 99% of patients were from Canadian and American sites. 

Reviewer #2:

1. The claim of this paper is to examine patient-reported and functional outcomes in highly and less-active older participants following distal radius fracture. Additionally the authors assessed the relationship between pre-injury level of physical activity and postoperative outcomes after casting, volar plating system, percutaneous pinning and external fixator placement. How significant is for the traumatology to know the pre-injury level of physical activity if the surgeon choose the percutaneous pinning or the external fixator methods?

Response: Our study demonstrates that traumatologists should be aware that highly-active patients who undergo percutaneous pinning have greater MHQ composite scores (including satisfaction and activities of daily living) compared to less-active patients; however, both percutaneous pinning and external fixation have very specific roles in the management of distal radius fractures (open fractures and significant comminution) related to pin site care, risk of infection, and non-rigid fixation. Regarding fixation and treatment methods, pre-injury level of activity should be used as a guide to help surgeons caring for these older patients. Despite chronological age, highly-active patients who wish to return to their normal activities in the shortest period of time should have the option to undergo surgery, preferably with volar locking plate. 

In the current literature there are main reasons in operating elder patient with isolated radius fractures. On the one hand they have bad soft tissue conditions on the other hand you have to avoid a long anesthesia period. The claims of the study are not properly placed in this context.

Response: Although many of the patients likely have osteoporotic bone, all patients had closed distal radius fractures. Despite the swelling and soft tissue injury, wound complications following distal radius fractures are fairly low; Sirnio et al reported an ~0.8% culture-positive wound infection among 881 distal radius fracture patients who underwent volar locking plate [1]. Percutaneous pinning and external fixation pin site infections occurred in about 25% of participants [2]. 

Regional block is often the anesthesia of choice when fixing distal radius fractures; this obviates the medical risks associated with general anesthesia in this older group. Weighing the risks of wound complications, anesthesia and duration of procedure, treatment of highly-active older patients with distal radius reduction and volar plating is a reasonable option. 

2. The date and analyses support the claims but in their conclusion the authors are saying casting is a suitable treatment for less active patients based on comparable patient-reported outcomes. But this decision making has to set in a context related to soft tissue damage or other contraindication for an operative treatment of distal radius fractures.

Response: Although the force required to cause the fracture would also cause significant soft tissue contusion, we did not have any open fractures in this series. The current study includes patients with closed distal radius fractures only; open wounds or substantial soft tissue compromise that necessitated surgery was not encountered. Despite edema, swelling, and soft tissue damage, few patients suffer wound healing complications following ORIF DRF with volar plating. 

3. Line 58: Volare plates cannot be used in all elderly patients. The above conclusion applies only to a selective patient population

Response: Agreed. We have clarified this.

Lines 106-108: “This has been changed to, “Because of the greater PROs, the early mobility and lower risk of hardware infection reported in the literature, volar plating is preferable to other treatments for highly-active patients who request and meet indications for surgery.”

4. Line 70: Casting is reserved for low-demand, less-active patients…. There are also active patients who simply do not want surgery and therefore also qualify for casting treatments.

Response: We have edited this.

Lines 121-124: “Casting is reserved for low-demand, less-active patients or those patients who do not wish to undergo surgery. Surgical reduction and fixation, typically with a volar locking plate system (VLPS) is utilized in high-demand, active patients.”

5. Line 99: The instability criteria listed are not complete. If this is wanted or there were no unstable radius fractures with termination of the palmar / dorsal joint lip, termination of the styloid ulnar process near the base, radioulnar dissociation, dorsal tilt of the peripheral fragment > 20 °, palmar tilt of the peripheral fragment > 20 °.

Response: The criteria listed in the manuscript was used for the trial and thus were reported. 

6. Unfortunately, no tables were included in the manuscript for the reviewer. The table titles could only be read under numbers 358-363.

Response: Tables are now included in the manuscript.

7. Line 228/229: I think in the discussion a gradation of the criteria regarding the therapy algorithm should be worked out, whereby the preoperative activity level is weighted less than the fracture pattern

Response: We agree the fracture pattern should be the most important factor; however, often times elderly patients who otherwise meet criteria for surgical intervention based on the previously listed criteria are not considered surgical candidates based on chronological age. The purpose of the study is to expand the way we think about suitability for wrist surgery among the elderly. 

8. Line 242/243: If you are already performing an operative intervention, you should aim at least for an exercise-stable osteosynthesis in order to be able to carry out a functional after-treatment. Otherwise the operational risk of complications is too great.

Response: Agreed, this is why we recommend volar plating over the other two methods of fixation. Fortunately, volar plating is more commonly used for fixation than some of the other methods. 

9. Line 256/257: Even if the risk of falls in active patients surely increases, supervision cannot be required as a prerequisite for their activities. That would cause costs to rise massively

Response: Agreed, complete supervision of all elderly activities would be costly. We advocate continued vigilance in identification and treatment of osteopenia and osteoporosis as falls in the setting of bone mineral deficiency leads to these fragility fractures. 

References:

1. Sirnio K, Flinkkila T, Vahakuopus M, Hurskainen A, Ohtonen P, Leppilahti J. Risk Factors for Complications After Volar Plate Fixation of Distal Radius Fractures. J Hand Surgery European Volume. 2019; 44: 456-461.

2. Chung KC, Malay S, Shauver MJ, Kim HM, for the WRIST group. Assessment of distal radius fracture complications among adults 60 years or older: A secondary analysis of the WRIST randomized clinical trial. JAMA Netw Open. 2(1):e187053, 2019.

---

## [Decision Letter · Decision Letter 1]

31 Mar 2020

PONE-D-20-03237R1

Pre-Injury Activity Predicts Outcomes Following Distal Radius Fractures in Patients Age 60 and Older

PLOS ONE

Dear Dr. Hooper,

Thank you for submitting your revised manuscript to PLOS ONE. After careful consideration, we feel that it has improved but does not fully meet the statistical requirements as it currently stands. Therefore, we invite you to submit a revised version of the manuscript that addresses the points raised during the statistical review process.

The comments of the staitistical reviewer are pointed out in detail. 

We would appreciate receiving your revised manuscript by may 31, 2020. To enhance the reproducibility of your results, we recommend that if applicable you deposit your laboratory protocols in protocols.io, where a protocol can be assigned its own identifier (DOI) such that it can be cited independently in the future. For instructions see: http://journals.plos.org/plosone/s/submission-guidelines#loc-laboratory-protocols

We look forward to receiving your revised manuscript.

Kind regards,

Hans-Peter Simmen, M.D., Professor of Surgery

Academic Editor

PLOS ONE

Reviewers' comments:

Reviewer's Responses to Questions

**Comments to the Author**

1. If the authors have adequately addressed your comments raised in a previous round of review and you feel that this manuscript is now acceptable for publication, you may indicate that here to bypass the “Comments to the Author” section, enter your conflict of interest statement in the “Confidential to Editor” section, and submit your "Accept" recommendation.

Reviewer #3: (No Response)

2. Is the manuscript technically sound, and do the data support the conclusions?

Reviewer #3: Partly

3. Has the statistical analysis been performed appropriately and rigorously? 

Reviewer #3: No

4. Have the authors made all data underlying the findings in their manuscript fully available?

Reviewer #3: Yes

5. Is the manuscript presented in an intelligible fashion and written in standard English?

Reviewer #3: Yes

6. Review Comments to the Author

Reviewer #3: Regarding the statistical methodology used in this manuscript, I have the following concerns:

Once a covariate has been detected as significant, please explain the interest of unadjusted comparisons between groups.

In this manuscript, an effect size measure is mentioned as if there were only one possible index, but there are several choices. To begin with, it must be highlighted that all the effect sizes are standardized ones, in opposition to the raw ones. On the other hand, there are different possibilities for these standardized measures. I am not sure if this manuscript reports Cohen's d or another related effect size. This must be properly stated. On the other hand, Cohen's d has been proved to be a biased standardized effect size and several corrections are available.

Moreover, regarding the effect sizes, I disagree with the idea of these measures are not applicable when a difference is lacking statistical significance. Precisely, the p-values are affected by the sample size while the standardized effect size measures are not. So, standardized effect sizes are a good guideline to assess the statistical power involved in a comparison. On the other hand, there are comparisons (eg table 3) with statistical significance where the effect size is not available.

Linear Mixed Models and GEE are different methodologies. The first one is not mentioned in the section of methodology but it is used to obtain the results in table 4. The election of each methodology must be properly explained.

Regarding table 4, I find weird to give one coefficient estimation of the model for each instant of measure. If the model considers time as a variable, there must be only one coefficient for it (considering the trend is linear), not one coefficient for each time value.

My view is that there is a high level of duplicity between the information given in tables and that which is given in the text.

Finally, decimal figures of p-values must be unified. APA's recommendation is to use three.

7. PLOS authors have the option to publish the peer review history of their article (what does this mean?). If published, this will include your full peer review and any attached files.

Reviewer #3: No

---

## [Author Response · Author response to Decision Letter 1]

13 Apr 2020

Reviewer #3: Regarding the statistical methodology used in this manuscript, I have the following concerns:

1. Once a covariate has been detected as significant, please explain the interest of unadjusted comparisons between groups.

We appreciate your comment. We intended to use the unadjusted differences to describe the imbalance between the two groups. However, it is not necessary to report the unadjusted p-values.

The modified table 3 has no unadjusted p-values. Formal comparisons are done by the controlled p-values.

2. In this manuscript, an effect size measure is mentioned as if there were only one possible index, but there are several choices. To begin with, it must be highlighted that all the effect sizes are standardized ones, in opposition to the raw ones. On the other hand, there are different possibilities for these standardized measures. I am not sure if this manuscript reports Cohen's d or another related effect size. This must be properly stated. On the other hand, Cohen's d has been proved to be a biased standardized effect size and several corrections are available.

Thank you for pointing this out. The effect size was measured using the standardized mean difference adapted from a paper by Kotsis et. al (https://www-sciencedirect-com.proxy.lib.umich.edu/science/article/pii/S0363502306010859) in a similar clinical setting.

We calculated the standardized mean difference in response using the formula: (mean response in highly active group – mean response in less active group)/SD of the measurement in two groups. It is used to measure the magnitude of difference between the two groups, which is different from Cohen’s d. Initially the Kotsis et. al uses ranges small (0.2-0.4), medium (0.50-0.70) and large (>0.80) as cutoffs. In our scenario, the effect sizes can be negative when less active group have better response. However, the absolute number of effect size specifies the magnitude of the difference clinically and could follow the same ranges. This is present in the manuscript, lines 152-156. 

3. Moreover, regarding the effect sizes, I disagree with the idea of these measures are not applicable when a difference is lacking statistical significance. Precisely, the p-values are affected by the sample size while the standardized effect size measures are not. So, standardized effect sizes are a good guideline to assess the statistical power involved in a comparison. On the other hand, there are comparisons (eg table 3) with statistical significance where the effect size is not available.

Thank you so much for your comment. We have added the effect sizes for non-significant parameters and unadjusted comparisons in tables 3 and 4. For table 3, if the estimators are not statistically different from 0, the effect sizes would be small. For table 4, the effect sizes examine the magnitude of the conditional mean difference at a specific time point. We used reference cell coding for time as a categorical variable (please find further explanation in point 5). 

4. Linear Mixed Models and GEE are different methodologies. The first one is not mentioned in the section of methodology but it is used to obtain the results in table 4. The election of each methodology must be properly explained.

We apologize. There was a typo in the label for table 4. We corrected the label as the “GEE model results”. The GEE model was preferred compared to the linear mixed model for its flexibility and allowing complicated correlation structures. Moreover, it only needs the 1st moment to be correctly specified rather than the whole likelihood. This has been updated in the manuscript, lines 163-168. 

5. Regarding table 4, I find weird to give one coefficient estimation of the model for each instant of measure. If the model considers time as a variable, there must be only one coefficient for it (considering the trend is linear), not one coefficient for each time value.

In the GEE models (table 4), we did not assume time to be linear. We used reference cell coding for time as a categorical variable, since we only have 4 time points in the data. 

The model for each patient outcome is specified as:

〖"E(Response" 〗_ij)=β_0+β_1 I("patient i is highly active " )+ β_2 I("patient i is highly active" )×I( 〖"time" 〗_ij="3 months") + β_3 I("patient i is highly active" )×I(〖"time" 〗_ij " = 6 months" )+β_4 I("patient is highly active" )×I(〖"time" 〗_ij " = 12 months" )+〖"demographics" 〗_i 

We have updated table 4 to better describe the model. The estimated mean differences are β_1(6 weeks), β_1+β_2 (3 months), β_1+β_3(6 months) and β_1+β_4 (12 months) respectively. 

Previously, we have reported p-values for β_1-β_4 in table 4. To better examine the group differences controlling for treatment received, time, within-patient correlation and other covariates, we have updated the p-values in table 4 with Wald test results testing H_a:β_1≠0,H_a: β_1+β_2≠0, H_a: β_1+β_3≠0 and H_a: β_1+β_3≠0 for time specific mean difference in response between highly active and less active groups. 

6. My view is that there is a high level of duplicity between the information given in tables and that which is given in the text.

Thank you for your comment. The tables provide all the statistical findings, we wanted to highlight some clinically important statistical findings from the tables in the text. This has been minimized. 

7. Finally, decimal figures of p-values must be unified. APA's recommendation is to use three.

Thank you for the suggestion. We have updated the tables 1-4 to unify 3 decimal figures for p-values.

---

## [Editor Report · Decision Letter 2]

21 Apr 2020

Pre-Injury Activity Predicts Outcomes Following Distal Radius Fractures in Patients Age 60 and Older

PONE-D-20-03237R2

Dear Dr. Hooper,

We are pleased to inform you that your manuscript has been judged scientifically suitable for publication and will be formally accepted for publication once it complies with all outstanding technical requirements.

With kind regards,

Hans-Peter Simmen, M.D., Professor of Surgery

Academic Editor

PLOS ONE
---

## [Editor Report · Acceptance letter]

28 Apr 2020

PONE-D-20-03237R2 

Pre-Injury Activity Predicts Outcomes Following Distal Radius Fractures in Patients Age 60 and Older 

Dear Dr. Hooper:

I am pleased to inform you that your manuscript has been deemed suitable for publication in PLOS ONE. Congratulations! Your manuscript is now with our production department. 

With kind regards,

on behalf of

Dr. Hans-Peter Simmen 

Academic Editor

PLOS ONE